# DexCanvas: Bridging Human Demonstrations and Robot Learning for Dexterous Manipulation

## Abstract

We present DexCanvas, a large-scale hybrid real-synthetic human manipulation dataset containing 7,000 hours of dexterous hand-object interactions seeded from 70 hours of real human demonstrations, organized across 21 fundamental manipulation types based on the Cutkosky taxonomy (Feix et al., 2016). Each entry combines synchronized multi-view RGB-D, high-precision mocap with MANO hand parameters, and per-frame contact points with *physically consistent* force profiles. Our real-to-sim pipeline uses reinforcement learning to train policies that control an actuated MANO hand in physics simulation, reproducing human demonstrations while discovering the underlying contact forces that generate the observed object motion. DexCanvas is the first manipulation dataset to combine large-scale real demonstrations, systematic skill coverage based on established taxonomies, and physics-validated contact annotations. The dataset can facilitate research in robotic manipulation learning, contact-rich control, and skill transfer across different hand morphologies.

## 1 Introduction

Dexterous manipulation with high-DoF anthropomorphic hands is fundamental to robot learning: it enables the most general form of object interaction and is essential for robots to achieve human-level autonomy in unstructured environments (Yu & Wang, 2022; Ozawa & Tahara, 2017). The field has witnessed rapid advancement along two dimensions: diverse learning paradigms including reinforcement learning for contact-rich control (Chen et al., 2024; 2023) and diffusion-based methods for handling multimodal action distributions (Weng et al., 2024; Wu et al., 2024), alongside dramatic scale expansion from task-specific models to billion-parameter foundation models (Wen et al., 2025; Kim et al., 2024; Zitkovich et al., 2023). However, current flagship manipulation systems predominantly rely on parallel-jaw grippers, while generalizable control of anthropomorphic hands remains limited to simulation or narrow real-world scenarios. This gap highlights an opportunity: to unlock the full potential of dexterous manipulation, we need large-scale datasets that capture diverse human manipulation strategies with physically accurate contact dynamics and force profiles, the crucial signals for learning robust dexterous control.

Building such datasets requires careful consideration of data sources and collection methodologies. The choice between robot-generated and human-sourced data presents fundamental tradeoffs for learning manipulation. Robot data through teleoperation (Arunachalam et al., 2022) provides direct demonstrations on target hardware but faces severe scalability challenges: low operational bandwidth, prohibitive costs, and, particularly for high-DoF dexterous hands, heavy cognitive load on operators that often produces unnatural finger kinematics (Rajaraman et al., 2020). Synthetic generation offers a cost-effective alternative, with recent methods producing millions of grasps through optimization (Wang et al., 2022; Zhang et al., 2024a) or billions through generative models (Ye et al., 2025), yet these approaches frequently yield biomechanically implausible grasps and remain tied to specific robot morphologies. Human demonstrations provide a morphology-agnostic alternative, but existing collection methods each carry limitations. Internet-scale video datasets (Hoque et al., 2025; Damen et al., 2018) offer breadth but lack systematic coverage and quality guarantees. Vision-based methods (Chao et al., 2021; Hampali et al., 2020; Qin et al., 2022) suffer from self-occlusion and noisy 3D pose estimation. Mocap systems (Fan et al., 2023; Taheri et al., 2020; Wang et al., 2024)

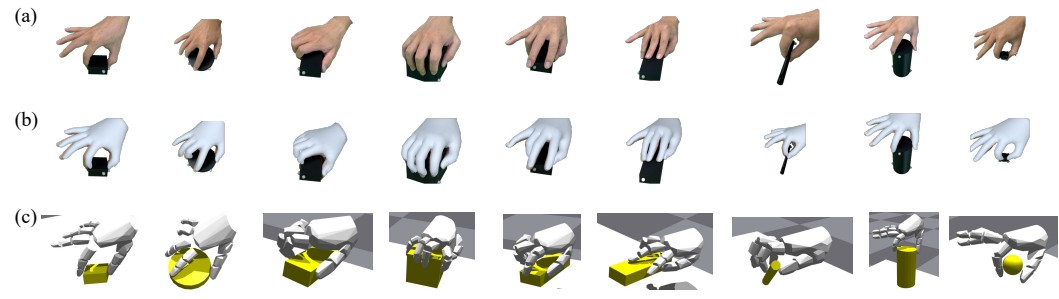

Figure 1: DexCanvas dataset overview. (a) Real human demonstrations captured through optical mocap showing diverse manipulation strategies. (b) MANO hand model fitted to mocap data preserving accurate kinematics. (c) Physics simulation with actuated MANO hand reproducing demonstrations while extracting contact forces and validating physical plausibility.

deliver precise kinematic trajectories but capture only geometric motion without the contact forces critical for manipulation.

We present DexCanvas[1], a large-scale hybrid real-synthetic dataset containing 7,000 hours of human manipulation data seeded from 70 hours of real demonstrations (Figure 1). Organized around 21 fundamental manipulation types derived from established grasp taxonomies (Feix et al., 2016; Chen et al., 2025), the dataset systematically covers strategies from power grasps to precision pinches and in-hand reorientations. The real component captures authentic human strategies through optical mocap with synchronized multi-view RGB-D. The synthetic component, generated through our real-to-sim pipeline, provides physically validated rollouts with complete force and contact annotations. Each entry includes MANO hand parameters (shape coefficients, joint angles, and keypoint coordinates), 6-DoF object poses, and physically consistent contact forces. This hybrid approach addresses the critical gaps in existing datasets: achieving both the scale needed for modern learning methods and the physical grounding essential for contact-rich manipulation.

Central to DexCanvas is our real-to-sim pipeline that transforms mocap demonstrations into physically validated data with force annotations. For each recorded manipulation, we train reinforcement learning (RL) policies to control an actuated MANO hand (Romero et al., 2017) in physics simulation, driving it to replicate the same object motion observed in the real demonstrations. The RL policy applies forces to reproduce the captured object trajectories while respecting physics constraints including friction, contact dynamics, and stability. From successful simulation rollouts, we extract force profiles directly from the physics engine: per-frame contact points, force vectors, and object wrenches that are exact up to simulation accuracy. This methodology transforms mocap from a geometry-only modality into a comprehensive source of manipulation data with both kinematic trajectories and the contact forces that produce them.

Our contributions are threefold: (1) We present DexCanvas, a hybrid real-synthetic dataset of 7,000 hours of human manipulation organized across 21 manipulation types, uniquely combining mocap trajectories, multi-view RGB-D, and physically consistent force profiles extracted through simulation. (2) We introduce a real-to-sim methodology using RL to extract contact forces from human demonstrations, transforming geometry-only mocap into complete manipulation data with force annotations—a technique applicable to existing mocap datasets. (3) We establish the foundation for cross-morphology transfer to diverse robot hands and release the complete dataset, preprocessing pipeline, and example code[2] to accelerate research in learning-based dexterous manipulation.

---

[1]Named as a foundational "canvas" for learning dexterous manipulation, where diverse human demonstrations serve as the raw material for training future robotic systems.

[2]Code and data are available at [redacted GitHub URL] and [redacted HuggingFace URL].

Table 1: Comparison with representative human hand-object interaction datasets.

| Dataset | Force/ Contact | Manip. Scope | Physics Valid | Scale (frames) | Modality | Annotation Source |
|---|---|---|---|---|---|---|
| *Vision-focused Datasets* | | | | | | |
| FreiHAND (Zimmermann et al., 2019) | ✗ | Poses | ✗ | 130K | RGB | Model fit |
| HO3D (Hampali et al., 2020) | ✗ | Poses | ✗ | 78K | RGB-D | Manual |
| DexYCB (Chao et al., 2021) | ✗ | Grasp | ✗ | 90K | RGB-D | Manual |
| HOI4D (Hampali et al., 2022) | ✗ | General | ✗ | 2.4M | RGB-D (ego) | Manual+auto |
| *Motion-focused Datasets* | | | | | | |
| GRAB (Taheri et al., 2020) | ✗ | Grasp | ✗ | 1.6M | Mocap | Markers |
| ARCTIC (Fan et al., 2023) | ✗ | Bimanual | ✗ | 2.1M | RGB-D+Mocap | Markers |
| Hi4D (Zheng et al., 2023) | Binary | Human* | ✗ | 11K | 4D scans | Multi-view |
| OpenEgo (Jawaid & Xiang, 2025) | ✗ | General | ✗ | 119.6M | Multimodal | Fusion† |
| EgoDex (Hoque et al., 2025) | ✗ | General | ✗ | 90M | RGB+3D | Vision Pro |
| *Synthetic/Hybrid Human Datasets* | | | | | | |
| ObMan (Hasson et al., 2019) | ✗ | Grasp | ✗ | 150K | Synthetic | Synthesis |
| RenderIH (Li et al., 2023) | ✗ | Two-hand | ✗ | 1M | Synthetic | Synthesis |
| HOIDiffusion (Zhang et al., 2024c) | ✗ | Grasp | ✗ | N/A‡ | Hybrid | Diffusion |
| *Contact-aware Datasets* | | | | | | |
| ContactDB (Brahmbhatt et al., 2019) | Binary | Grasp | ✗ | 375K | Thermal | Thermal |
| ContactPose (Brahmbhatt et al., 2020) | Binary | Grasp | ✗ | 2.9M | RGB-D+Thermal | Thermal |
| **DexCanvas (Ours)** | **Continuous** | **General** | ✓ | **3.0B** | RGB-D+Mocap | Simulation |

*Human-human interaction. †Unified from 6 datasets. ‡Generation method, not fixed dataset. *Manip. Scope*: Poses = static hand poses; Grasp = grasping actions; General = diverse manipulation tasks; Bimanual = two-handed manipulation; Two-hand = two hands interacting.

## 2 RELATED WORK

We first review existing hand-object interaction datasets to position DexCanvas within the current landscape. We then examine approaches to contact and force estimation, highlighting the methodological gap that our physics-based reconstruction addresses. Extended discussion of individual datasets and force estimation methods can be found in Appendix A.2.

### 2.1 HUMAN HAND-OBJECT INTERACTION DATASETS

Human demonstrations offer natural manipulation strategies but face fundamental annotation challenges. Human hand datasets reflect a progression of sensing capabilities (Table 1): early vision-based work focused on hand pose estimation (Zimmermann et al., 2019) or simple grasping (Chao et al., 2021), constrained by manual annotation costs. Motion capture systems (Taheri et al., 2020; Fan et al., 2023) achieved kinematic precision but captured only geometric motion. Recent multi-modal approaches scale dramatically with EgoDex (Hoque et al., 2025) and OpenEgo (Jawaid & Xiang, 2025) reaching 90M+ frames, yet none provide force measurements essential for contact-rich control. Binary contact from thermal imaging (Brahmbhatt et al., 2020) or 4D scanning (Zheng et al., 2023) partially addresses this gap but lacks force magnitudes. Synthetic generation (Hasson et al., 2019; Zhang et al., 2024c) offers perfect annotations but sacrifices naturalness. DexCanvas uniquely combines large-scale real human demonstrations with physics-simulated force profiles, preserving authentic motion while adding complete physical annotations.

### 2.2 SYNTHETIC ROBOT MANIPULATION DATASETS

While human datasets capture natural strategies, robot-specific synthetic datasets offer complementary advantages: unlimited scale and perfect ground truth. Synthetic datasets have evolved rapidly from early physics-validated collections (Aktaş et al., 2022) to billion-scale demonstrations (Ye et al., 2025), using either optimization-based methods (Wang et al., 2022; Zhang et al., 2024b) or generative models. The field now encompasses articulated object manipulation (Bao et al., 2023), bimanual coordination (Chen et al., 2023), and cluttered scenes (Zhang et al., 2024a). However, these datasets remain tied to specific robot morphologies and often lack natural adaptability. DexCanvas takes a hybrid approach: synthesizing from human demonstrations provides efficient real-world seeds encoding successful strategies, while serving as a morphology-agnostic bridge for retargeting to diverse robot hands.

## 2.3 CONTACT AND FORCE ESTIMATION

Current approaches to force annotation face a fundamental dilemma: direct measurement requires instrumented objects that constrain natural manipulation, while indirect methods provide only approximate estimates. Vision-based approaches (Pham et al., 2015; Grady et al., 2022) lack ground truth validation, contact-aware reconstruction (Corona et al., 2020; Hasson et al., 2019; Zhou et al., 2022) cannot measure actual forces, and thermal imaging (Brahmbhatt et al., 2019) provides only binary contact masks. DexCanvas addresses this through physics-based reconstruction: training RL policies to reproduce human demonstrations in simulation extracts physically consistent force profiles from observed kinematics, preserving natural motion while providing force annotations grounded in physics.

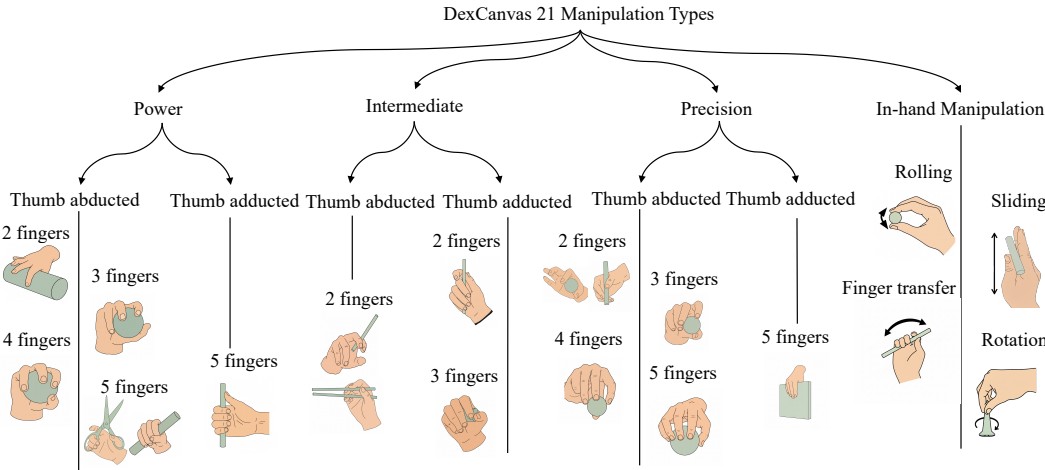

Figure 2: Complete taxonomy of 21 manipulation types in DexCanvas. The taxonomy hierarchically organizes manipulation strategies into four main categories: Power (whole-hand grasps for stability), Intermediate (transitional grasps combining power and precision), Precision (fingertip control for dexterity), and In-hand Manipulation (dynamic object reorientation). Each category is further subdivided by thumb position and finger participation, with visual demonstrations showing the characteristic hand configurations for each manipulation type.

## 3 DATASET CONSTRUCTION AND PROCESSING PIPELINE

DexCanvas is a large-scale dataset containing 7,000 hours of human dexterous manipulation, combining 70 hours of real mocap demonstrations with physics-simulated expansions that provide contact force annotations. This section describes how we construct the dataset: the manipulation taxonomy and collection protocol that guides what data to capture (§3.1), the multi-modal hardware system used to record human demonstrations (§3.2), and the processing pipeline that transforms raw captures into standardized MANO representations and synthesizes force profiles through physics simulation (§3.3). Together, these components enable DexCanvas to provide both authentic human manipulation strategies and the physical annotations essential for learning contact-rich control.

### 3.1 MANIPULATION TAXONOMY AND DATA COLLECTION DESIGN

We organize DexCanvas around 21 fundamental manipulation types, systematically derived from the Cutkosky taxonomy (Feix et al., 2016). These are grouped into four major categories: precision grasps, power grasps, intermediate grasps, and in-hand manipulations (rolling, sliding, finger transfer, and rotation). Figure 2 illustrates the complete taxonomy with visual examples of each manipulation type, organized by thumb position (abducted/adducted) and number of participating fingers. This organization ensures systematic coverage of dexterous human capabilities, moving beyond ad-hoc task-specific motions to capture the breadth of manipulation strategies relevant for robotic learning.

**Object selection:** Our 30-object set includes geometric primitives in multiple sizes to test scale adaptation, weight-varied duplicates (marked 'H') to probe force modulation, and YCB objects (Chao et al., 2021) like the power drill and pitcher for complex task-specific grasps. See Appendix A.3.1.

**Data collection:** Five operators performed 50 repetitions of each feasible manipulation–object pair after training on standardized demonstrations. Trials followed a consistent structure: object placement, manipulation execution, and return to neutral. In total, we collected 12,000 sequences spanning 70 hours of demonstrations, excluding trials with drops or significant occlusions.

## 3.2 MULTI-MODAL CAPTURE SYSTEM

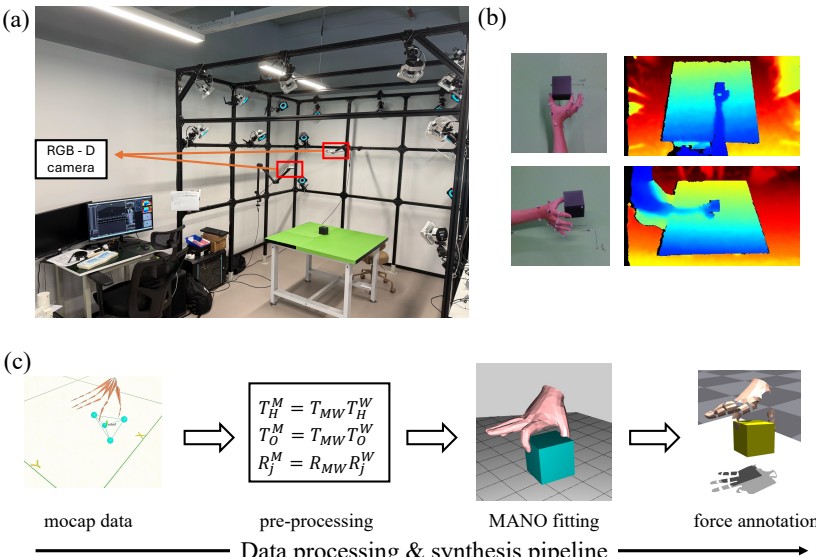

Figure 3: Overview of the capture setup and data pipeline. (a) The room-scale optical motion capture system with 22 infrared cameras and synchronized RGB-D sensors. (b) Samples from the two calibrated RGB-D cameras: top row shows RGB view and depth map from the front perspectives and the bottom row shows the bottom perspective. (c) Data processing pipeline. Raw mocap markers are pre-processed and transformed into the MANO coordinate system, aligned with RGB-D observations, and fitted to the MANO hand model. The resulting hand–object trajectories are then used to train reinforcement learning policies in simulation.

Our capture system employs 22 infrared cameras for millimeter-precision optical mocap alongside two synchronized RGB-D cameras (Fig. 3(a-c)).

**Marker instrumentation:** We attach 14 reflective markers to the right hand at anatomically meaningful locations and 4 markers to each object. The key insight is that objects are 3D-printed from CAD models with marker mounting locations carved directly into the geometry. This eliminates coordinate frame misalignment and the tracked pose corresponds exactly to the URDF model used in physics simulation.

**Data streams:** The system captures hand pose (6-DoF wrist, finger joints), object trajectories, and multi-view RGB-D at 30 Hz. Raw marker positions are processed to extract joint angles, with brief occlusions handled through interpolation. Per-participant calibration accounts for hand size variations, while time synchronization ensures frame-level correspondence across all modalities.

## 3.3 PROCESSING AND SYNTHESIS PIPELINE

**MANO shape:** Raw mocap markers are processed into MANO representations (Romero et al., 2017). We first estimate per-participant shape parameters ($\beta \in \mathbb{R}^{10}$) from calibration sequences,

then optimize frame-wise wrist transforms and joint angles to minimize marker-to-surface distances while respecting anatomical constraints (details in Appendix A.3.2).

**Data processing:** As illustrated in Figure 3(c), the mocap system produces raw hand/object pose's trajectories, which are transformed into the MANO coordinate system through a pre-processing pipeline (Appendix A.4). The processed parameters are then passed through the MANO forward model to reconstruct the hand mesh and joint locations, enabling consistent visualization and downstream use in simulation.

**Physics-based force extraction:** The key innovation is using RL to discover contact forces from kinematics alone. We train policies to control an actuated MANO hand in IsaacGym (Makoviychuk et al., 2021), reproducing captured object motion while physics simulation reveals the underlying forces. Successful rollouts provide per-frame contact points, force vectors, and torques that are physically consistent with the observed motion. This approach transforms 70 hours of mocap into 7,000 hours with complete force annotations. Section 4 details the RL methodology.

## 4 PHYSICS-BASED FORCE RECONSTRUCTION VIA RL

The core challenge in extracting forces from mocap is that geometry alone cannot determine contact dynamics: even millimeter-accurate tracking contains systematic errors that lead to penetrations or loss of contact when replayed open-loop in simulation. Our solution (Figure 4) uses reinforcement learning to train closed-loop tracking controllers that reproduce the observed object motion while the physics simulator provides ground-truth force measurements.

The key insight is that the RL policy acts as a residual controller, adding small corrections to the fitted MANO joint angles to maintain stable contact. The policy observes the complete mocap data—hand kinematics, object trajectory, and future object poses—and outputs joint angle residuals that compensate for tracking errors. Crucially, the forces are not inferred by the policy but directly measured by the physics simulator during rollout. This approach transforms the force reconstruction problem into a tracking control problem where physical consistency is enforced by the simulator. Beyond annotation, policy rollouts enable data synthesis by perturbing object sizes, initial poses, material properties, and MANO shape parameters, generating diverse physically valid variations from each demonstration.

### 4.1 PROBLEM SETUP

We train one policy $\pi_\theta$ for each object-manipulation pair, avoiding the complexity of multi-task conditioning. Each manipulation is modeled as a Markov Decision Process (MDP) $\mathcal{M} = (\mathcal{S}, \mathcal{A}, \mathcal{P}, R, \gamma)$, where the state $s_t \in \mathcal{S}$ includes both hand and object kinematics. The action $a_t \in \mathcal{A} \subset \mathbb{R}^n$ specifies continuous joint residuals added to the fitted MANO angles. $\mathcal{P}$ represents the physics-based transition dynamics, $R$ is the reward function, $\gamma$ is the discount factor, and episodes terminate on success or failure within horizon $T$. The policy is a stochastic controller:

$$a_t \sim \pi_\theta(a_t \mid s_t). \tag{1}$$

The objective is to maximize the expected discounted return:

$$\max_\theta \ J(\theta) = \mathbb{E}_{\pi_\theta} \left[ \sum_{t=0}^{T} \gamma^t R(s_t, a_t) \right]. \tag{2}$$

The reward structure implements a dual objective: accurate object trajectory tracking (following Chen et al. (2024)) and minimal deviation from the original manipulation gesture through residual penalties. This formulation ensures the policy reproduces the demonstrated object motion while maintaining fidelity to human hand kinematics.

For each object-manipulation pair, we load the time-aligned demonstrations in MANO format, providing reference hand poses and object trajectories. The policy learns residual corrections to these poses, outputting joint angle adjustments that maintain stable contact. After training with PPO (Schulman et al., 2017), we roll out the policy and retain only physically valid trajectories. These rollouts provide per-frame contact points, force vectors, and torques directly measured by the physics simulator which information impossible to obtain from mocap alone.

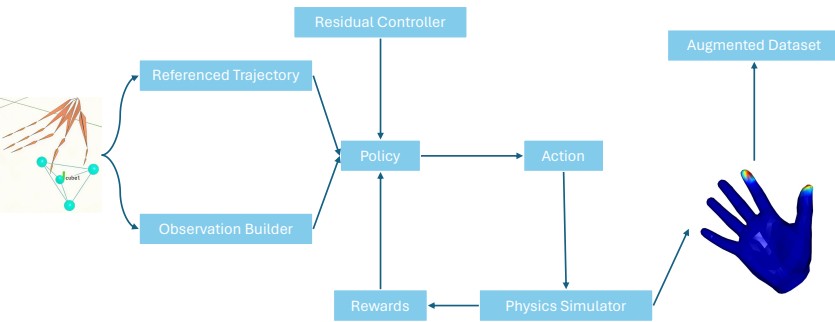

Figure 4: Overview of the reinforcement learning–based force reconstruction pipeline.

## 4.2 TRAINING

**Data loading and initialization.** Training uses processed mocap data transformed into MANO coordinates. At initialization, the MANO hand matches the captured configuration while the object follows its mocap pose. The simulator tracks object trajectory errors for reward computation.

**Action space.** The MANO hand operates as a floating-base system with actuated fingers. Actions are residual corrections to fitted joint angles, not absolute commands. These residuals accumulate through exponential filtering, keeping the policy close to human demonstrations while compensating for tracking errors (Appendix A.5.1).

**Observation space.** The policy observes complete non-causal mocap data including future object poses—privileged information that improves tracking without requiring real-world deployment. During evaluation, we log contact points and forces from the simulator (Appendix A.5.2).

**Reward function.** The reward balances object trajectory tracking with gesture fidelity through residual penalties. This ensures physically valid motion that preserves human kinematics. Episodes terminate upon trajectory completion or excessive divergence. Training uses PPO with detailed reward decomposition in Appendix A.5.3.

## 5 EXPERIMENTS

**Methodology effectiveness and data synthesis.** To validate our processing pipeline, we trained policies for all feasible object–manipulation pairs and evaluated success rates on 32 representative pairs shown in Figure 5. Each policy was rolled out 100 times in simulation, with success defined as completing the demonstration trajectory without termination (triggered when positional error exceeded 5cm).

Policies achieved an 80.15% success rate under nominal conditions, demonstrating effective reproduction of human demonstrations in physics simulation. When initial object poses were perturbed by up to 20% of object size, the success rate decreased moderately to 62.54%—a drop of only 17.61 percentage points. This moderate degradation under substantial perturbations shows that each policy can generate diverse, physically valid training data from a single demonstration seed, significantly expanding dataset coverage. Our synthesis pipeline thus transforms 70 hours of real demonstrations into 7,000 hours of physics-validated rollouts—a 100× expansion critical for scaling to modern learning requirements.

**Force annotation quality.** We next evaluate the quality of the contact force annotations produced by our real-to-sim pipeline.

Figure 6 illustrates an example manipulation. Figure 6(a) shows the input trajectory reproduced in simulation. Figure 6(b) plots the per-finger contact force magnitudes over time, together with the maximum force at each timestep. Figure 6(c) visualizes the rendered force distributions on the hand mesh at selected keyframes.

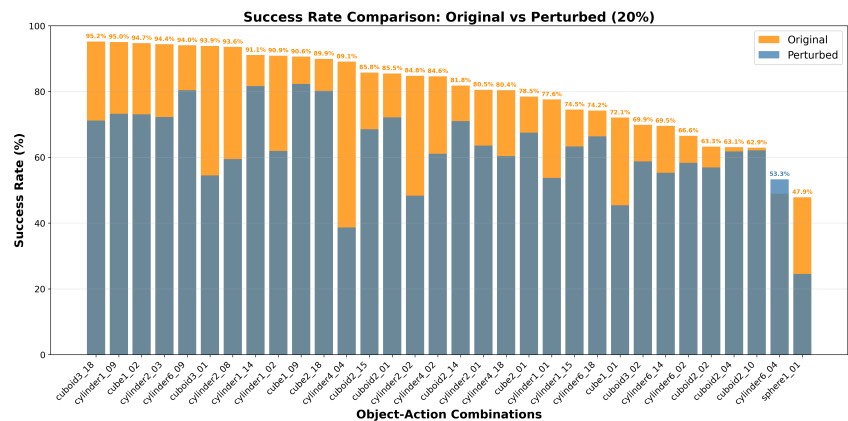

Figure 5: Success rates of policies across 32 representative object–manipulation pairs under both nominal and perturbed initial conditions. Bars show success under perturbed settings (20% of object size shift in length and width), with orange overlays indicating the success rates achieved under the original conditions.

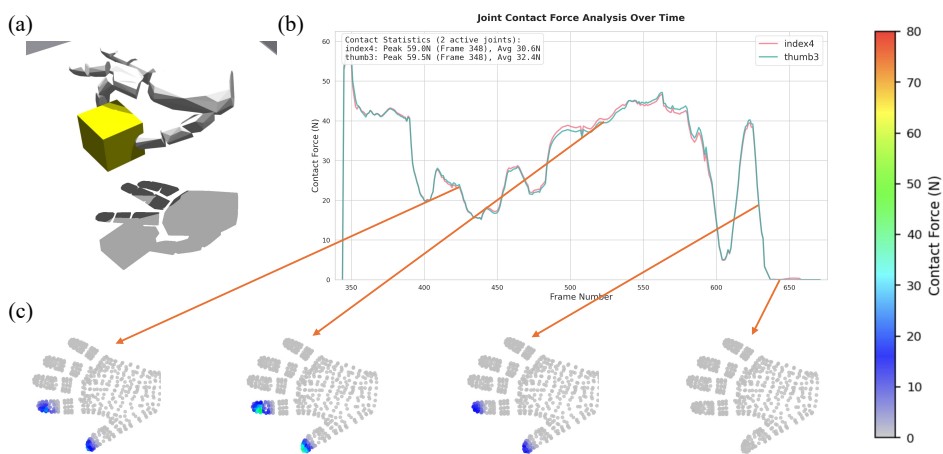

Figure 6: Evaluation of force annotation quality. (a) Example manipulation trajectory reproduced in simulation. (b) Time series of per-finger contact forces and maximum contact force. (c) Rendered force distributions on the hand mesh at selected timesteps.

The temporal profiles in Figure 6(b) reveal smooth and physically consistent variations of contact force across different fingers, while the spatial renderings in Figure 6(c) highlight the correspondence between force peaks and the actual contact regions involved in the manipulation. Together, these results demonstrate that our reconstruction pipeline provides not only successful trajectory reproduction but also rich, fine-grained force annotations at the level of individual fingers. This level of annotation is rarely available in existing datasets and is positioned to provide valuable supervisory signals for learning contact-aware dexterous manipulation policies.

**Manipulation-specific force distributions.** We analyzed how different manipulation types from our taxonomy produce distinct contact force patterns. For three representative manipulation types, we aggregated contact statistics across 100 simulation rollouts to characterize their force signatures.

Figure 7(b) reveals distinct force signatures for each manipulation type: power grasps engage multiple joints uniformly while precision manipulations concentrate forces on specific fingertips. These characteristic patterns across our 21-type taxonomy demonstrate that our pipeline successfully captures the unique physical signatures of each manipulation strategy. The variation within each type also indicates our synthesis generates diverse valid executions, providing rich supervisory signals for learning manipulation-specific control policies.

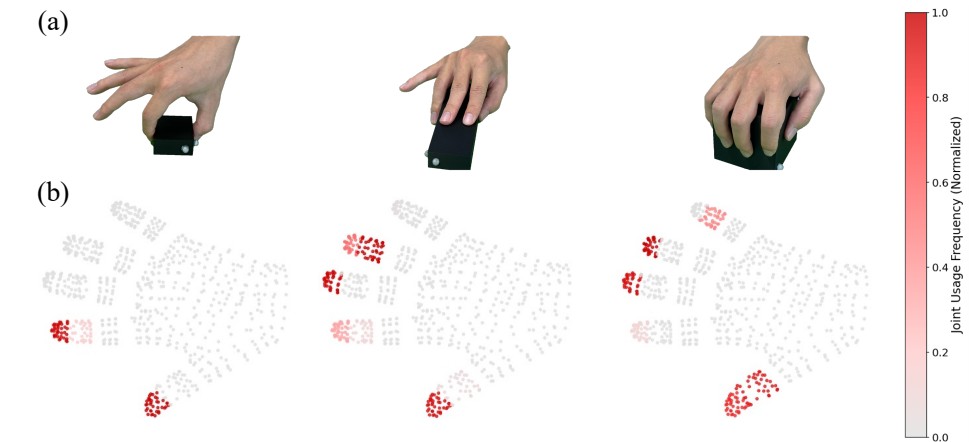

Figure 7: Manipulation-specific contact distributions. (a) Three representative manipulation types from our taxonomy. (b) Heatmaps of joint-level contact frequencies aggregated over 100 trials, revealing distinct force patterns for each manipulation type.

Additional experiments demonstrating cross-dataset applicability and downstream task evaluation will be provided during the rebuttal phase.

## 6 DISCUSSION AND CONCLUSION

DexCanvas provides a foundation for learning dexterous manipulation through human demonstrations augmented with physics-based force annotations. We acknowledge several limitations and outline concrete directions for future development.

**Expanding manipulation scope.** Our current coverage is limited to basic geometric objects and fundamental manipulation primitives. These serve as building blocks for more complex behaviors: from simple geometries to YCB objects to real-world assets, and from basic grasps through in-hand reorientations to free-form scenarios combining multiple primitives.

**Unified policy learning.** Our per-object-manipulation policy training faces obvious scalability challenges. A unified MANO control model that conditions on object and task encodings could reproduce diverse trajectories in bulk, eliminating thousands of specialized policies. This becomes critical for longer sequences where skills must compose naturally.

**Cross-morphology retargeting.** The dataset currently releases only human hand data, though our methodology provides the foundation for successful retargeting to robot morphologies. The force annotations enable physics-aware retargeting that preserves contact dynamics across different hand designs, from underactuated to fully anthropomorphic systems, while maintaining physical consistency with human demonstrations.

**Multi-modal annotations.** RGB-D streams are included but unexplored. The visual modality enables visuomotor policy learning, while photorealistic rendering or world-model-based synthesis could generate unlimited perception training data. Language annotations remain minimal, but force measurements provide physical grounding: "gentle" versus "firm" grasps correspond to measurable force profiles rather than subjective descriptions.

**Downstream applications.** The dataset supports diverse research directions. For reinforcement learning, force annotations enable reward shaping and contact-aware exploration. For vision-language-action model pretraining, the combination of visual observations, manipulation primitives, and physical measurements provides rich multi-modal supervision. By releasing DexCanvas with complete preprocessing pipelines and baseline implementations, we aim to accelerate progress toward capable robotic manipulation systems.

## ETHICS STATEMENT

The DexCanvas dataset was collected following institutional guidelines with informed consent from all human demonstrators. Participants were compensated fairly for their time and could withdraw from the study at any point. All personally identifiable information has been removed from the released data, including face regions in RGB images and any identifying markers. The dataset contains only manipulation of inanimate objects with no sensitive content. While the dataset could theoretically enable more capable manipulation systems, we believe the benefits for advancing robotic assistance and automation outweigh potential risks, particularly given the focus on fundamental manipulation primitives rather than task-specific capabilities.

## REPRODUCIBILITY STATEMENT

To ensure reproducibility of our results, we provide comprehensive implementation details throughout the paper and appendices. The complete data processing pipeline is described in Section 3.3 and Appendix A.3, with mathematical formulations for MANO fitting and force reconstruction. The RL training methodology (Section 4) includes full specifications of state/action spaces, reward functions, and hyperparameters in Appendices A.4.1-A.4.3. All code for data preprocessing, physics simulation setup, and policy training will be released at [redacted GitHub URL]. The dataset itself, including raw mocap, RGB-D streams, and synthesized force annotations, will be available at [redacted HuggingFace URL] with documented data formats and loading utilities. Hardware specifications for the motion capture system are detailed in Section 3.2. The physics simulation uses IsaacGym with specified parameters for contact modeling and friction coefficients. Trained policies and checkpoints will be released to enable direct replication of force extraction results.

## LLM USAGE STATEMENT

Large language models played a significant role in this work beyond simple writing assistance. LLMs were used to: brainstorm research directions and technical approaches, organize and synthesize related literature, implement data processing pipelines, develop the reinforcement learning framework for force extraction, and write experimental code. However, the following were NOT generated by LLMs: (1) the core research idea of combining human demonstrations with physics-based force reconstruction, (2) the strategic positioning of the dataset within the manipulation landscape, (3) the actual dataset collection and human demonstrations, and (4) all reported experimental results and quantitative metrics. The authors have carefully verified all technical claims and factual statements in this paper to ensure accuracy and prevent hallucination artifacts.

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

# A APPENDIX

## A.1 DATASET ACCESS AND USAGE

DexCanvas is hosted on HuggingFace Hub and supports both full download and streaming access. The dataset provides two fundamentally different access patterns optimized for supervised and reinforcement learning paradigms.

### A.1.1 DATA FORMAT

Each trajectory contains metadata and synchronized multi-modal sequences. The metadata specifies the demonstrator, manipulated object, and manipulation type from the Cutkosky taxonomy, along with frame ranges marking active manipulation periods and subject-specific MANO shape parameters. Sequences include MANO hand kinematics (wrist pose and 45D finger joints), 6-DoF object poses, and physics-simulated contact information with per-frame force vectors and object wrenches.

Table 2: DexCanvas trajectory data structure

| Category | Field | Dimensions |
|---|---|---|
| *Metadata* | | |
| | operator | String ID |
| | object | String ID |
| | manipulation_type | Cutkosky category |
| | mano_shape | [10] |
| | active_frames | Start/end indices |
| | fps | 30Hz (RGB-D), 120Hz (mocap/sim) |
| *Hand kinematics* | | |
| | wrist_position | [T, 3] |
| | wrist_rotation | [T, 3] |
| | finger_pose | [T, 45] |
| *Object trajectory* | | |
| | object_position | [T, 3] |
| | object_rotation | [T, 3] |
| *Contact forces* | | |
| | contact_points | [T, N, 3] |
| | force_vectors | [T, N, 3] |
| | object_wrench | [T, 6] |

All modalities are temporally aligned with RGB-D captured at 30 Hz while mocap and physics simulation run at 120 Hz, providing 4× temporal resolution for precise contact dynamics.

### A.1.2 SUPERVISED LEARNING ACCESS

For trajectory prediction, behavior cloning, and motion analysis, the dataset provides standard PyTorch DataLoader compatibility:

```python
from datasets import load_dataset
from torch.utils.data import DataLoader

# Stream directly from HuggingFace Hub without downloading
dataset = load_dataset("[ANONYMIZED]/DexCanvas", streaming=True)

# Filter by manipulation type, object, or demonstrator
filtered = dataset.filter(
    lambda x: x.manipulation_type in ["tripod", "palmar_pinch"] and
              x.object in ["cube2", "sphere1"]
)

# Standard batch-synchronous iteration
for batch in DataLoader(filtered, batch_size=32):
```

```
mano_poses = batch["finger_pose"]      # [B, T, 45]
object_poses = batch["object_info"]    # [B, T, 6]
forces = batch["contact_info"]         # [B, T, ...]
masks = batch["attention_mask"]        # [B, T] for variable lengths
```

### A.1.3 REINFORCEMENT LEARNING ACCESS

For parallel RL training, the dataset provides a stateful trajectory buffer with fundamentally different characteristics. Unlike batch-synchronous processing, each parallel environment maintains persistent GPU storage of its assigned trajectory and tracks its own timestep independently. When environment $i$ completes its episode, only trajectory $i$ is replaced while other environments continue uninterrupted. Contact forces and wrenches remain GPU-resident for direct reward computation without CPU transfers. This asynchronous design enables scaling to thousands of parallel physics environments where each requires independent, stateful trajectory playback.

```python
import torch
from dexcanvas import DexCanvasDataset, RLTrajectoryBuffer

# Initialize stateful buffer for parallel environments
dataset = DexCanvasDataset(manipulation_types=["tripod", "pinch"])
buffer = RLTrajectoryBuffer(dataset, num_envs=1024, device="cuda")

# Each environment tracks its own timestep
for step in range(max_steps):
    # Get current observations for all environments
    obs = buffer.get_observations()  # Zero-copy GPU indexing
    hand_poses = obs["mano_pose"]    # [num_envs, 45]
    object_poses = obs["object_pose"] # [num_envs, 6]
    contact_forces = obs["forces"]   # [num_envs, N, 3]

    # Step environments (physics simulation)
    actions = policy(obs)
    rewards = compute_rewards(contact_forces, object_poses)
    dones = check_termination()

    # Asynchronously replace completed trajectories
    if dones.any():
        done_envs = torch.where(dones)[0]
        buffer.reset_envs(done_envs)  # Only these envs get new data
    else:
        buffer.advance_timesteps()     # Others continue current
            trajectory
```

### A.1.4 FLEXIBLE DATA SELECTION

Researchers can filter trajectories along multiple dimensions. The 21 manipulation types from the Cutkosky taxonomy enable targeted experiments on specific strategies like power grasps, precision pinches, or in-hand rotations. Geometric primitives come in multiple size and weight variants—heavy objects marked with 'H' provide contrastive force profiles for studying adaptation to object properties while maintaining identical manipulation strategies. Individual demonstrator IDs allow analysis of personal styles and hand morphology variations. The complete dataset contains 3.0 billion frames with consistent MANO parameterization and physics-validated force annotations, enabling both learning from human demonstrations and systematic evaluation of contact-rich manipulation strategies.

## A.2 EXTENDED RELATED WORKS

### A.2.1 EVOLUTION OF HUMAN HAND-OBJECT INTERACTION DATASETS

The progression of human manipulation datasets reflects evolving sensing capabilities and research priorities. Early vision-based work like FreiHAND (Zimmermann et al., 2019) established large-

scale collection methodologies but remained limited to static hand poses, missing the dynamics essential for understanding manipulation. When researchers shifted focus to actual object interaction, fundamental challenges emerged: HO3D (Hampali et al., 2020) and DexYCB (Chao et al., 2021) achieved multi-view capture but required months of manual annotation for relatively small datasets, revealing the scalability bottleneck of vision-only approaches.

Motion capture systems promised to overcome these limitations through millimeter-precision tracking. GRAB (Taheri et al., 2020) demonstrated that mocap could capture natural whole-body grasping behaviors, while ARCTIC (Fan et al., 2023) extended this to complex bimanual manipulation with synchronized RGB-D streams. However, even these high-precision systems exposed a critical gap: they capture geometric motion but not the forces driving that motion. This limitation becomes particularly acute when attempting to transfer human demonstrations to robots, where force regulation determines success or failure.

The recent emergence of large-scale egocentric datasets marks a paradigm shift in how we capture human manipulation. EgoDex (Hoque et al., 2025), collected using Apple Vision Pro, provides 829 hours of tabletop manipulation with native 3D hand tracking—a capability absent from earlier egocentric datasets like Ego4D. OpenEgo (Jawaid & Xiang, 2025) takes a complementary approach by unifying six existing datasets into a coherent framework with language-aligned action primitives, addressing the fragmentation that has plagued the field. These datasets achieve unprecedented scale but inherit the fundamental limitation of all observational data: they cannot measure the forces that produce the observed motion.

Synthetic generation offered a potential solution to both scale and annotation challenges. Ob-Man (Hasson et al., 2019) demonstrated that optimization could produce unlimited hand-object interactions with perfect geometric annotations, yet the resulting grasps often violated biomechanical constraints. ContactPose (Brahmbhatt et al., 2020) attempted to bridge the real-synthetic gap through thermal imaging, capturing heat signatures at contact points. While innovative, thermal imaging provides only binary contact masks and suffers from proximity artifacts where heat transfer occurs without actual contact. The field needed a method to extract continuous force profiles from natural human demonstrations.

### A.2.2 Synthetic Robot Manipulation Datasets: From Optimization to Generation

The evolution of synthetic manipulation datasets reveals a fundamental tension between physical validity and scale. Early work like DexGraspNet (Wang et al., 2022) employed differentiable force closure optimization to ensure stable grasps, producing 1.32 million physically valid demonstrations. However, optimization-based approaches faced an inherent bottleneck: each grasp required expensive iterative computation, limiting both scale and diversity.

This limitation drove the field toward generative approaches. DexGraspNet 2.0 (Zhang et al., 2024a) introduced a hybrid pipeline where optimization creates seed data that trains conditional diffusion models, achieving a 300-fold increase to 427 million grasps while maintaining 90.7% real-world success rates. The key insight was that generative models could learn the manifold of valid grasps from optimization examples, then rapidly sample new instances without explicit physics computation.

Dex1B (Ye et al., 2025) pushed this paradigm to its logical extreme with one billion demonstrations across multiple hand morphologies. Rather than treating each hand design as requiring separate datasets, Dex1B trains unified models that generate grasps for Shadow, Inspire, and Ability hands. The dataset employs a conditional VAE with geometric constraints, using approximate signed distance functions to prevent interpenetration while maintaining computational efficiency. This approach reveals that the bottleneck has shifted from data generation to data utilization—we can now produce more synthetic demonstrations than current learning algorithms can effectively consume.

Specialized datasets have emerged to address specific manipulation challenges beyond grasping. BiDexHands (Chen et al., 2023) focuses on bimanual coordination, achieving 40,000+ FPS simulation to enable reinforcement learning at scale. DexArt (Bao et al., 2023) tackles articulated object manipulation, where success requires reasoning about both hand and object kinematics. Robo-Casa (Nasiriany et al., 2024) takes a task-centric approach with 100 evaluation scenarios in kitchen environments, using large language models to generate composite task specifications. These spe-

cialized datasets highlight that scale alone is insufficient—the field needs diverse task coverage and evaluation protocols.

### A.2.3 THE FORCE ESTIMATION CHALLENGE: FROM VISION TO PHYSICS

The quest to estimate manipulation forces from observation has produced increasingly sophisticated methods, each revealing new aspects of why this problem resists simple solutions. Pham et al. (2015) established the theoretical framework by formulating force estimation as an inverse dynamics problem: given observed object motion, what forces must the hand apply? Their approach combined visual tracking with second-order cone optimization, discovering that humans consistently apply "excessive forces" beyond mechanical requirements—a finding that highlights the gap between theoretical force minimization and natural manipulation.

Modern learning approaches shifted from optimization to direct regression. PressureVision (Grady et al., 2022) exploits subtle visual cues that correlate with applied pressure: soft tissue deformation, blood flow changes, and cast shadows. By training on controlled recordings where participants pressed on instrumented surfaces, the system learns to map appearance changes to pressure distributions. Yet without ground truth forces during deployment, validation remains confined to qualitative assessment and controlled scenarios.

Contact-aware reconstruction methods took a different path, focusing on geometric consistency rather than force measurement. GanHand (Corona et al., 2020) uses adversarial training to ensure plausible hand-object configurations, while TOCH (Zhou et al., 2022) maintains temporal consistency through learned motion priors. These methods produce visually convincing results but cannot distinguish between gentle placement and forceful grasping—geometrically identical contacts can involve vastly different force profiles.

Direct measurement through instrumented objects or thermal imaging seemed to offer ground truth, but each approach introduced new limitations. Force-torque sensors alter the contact mechanics they aim to measure, while thermal cameras capture only contact presence without force magnitude. ContactDB (Brahmbhatt et al., 2019) demonstrated that even with specialized sensors, capturing distributed contact patterns across the entire hand surface remains infeasible. This progression of methods reveals a fundamental insight: force estimation from observation alone is under-constrained, requiring either physical sensors that disrupt natural behavior or physics simulation that can enforce consistency between motion and forces.

### A.2.4 MANIPULATION TAXONOMY: THE GAP BETWEEN THEORY AND PRACTICE

The disconnect between theoretical grasp taxonomies and dataset coverage reveals how the field has prioritized certain aspects of manipulation while neglecting others. The Cutkosky taxonomy's 16 grasp types and the GRASP taxonomy's expansion to 33 types (Feix et al., 2016) provide comprehensive frameworks for categorizing human manipulation strategies. Yet most datasets capture fewer than five grasp types, typically focusing on power grasps and simple pinches while ignoring sophisticated patterns like finger gaiting or in-hand reorientation.

This coverage gap stems partly from collection challenges—dynamic manipulations are harder to capture and annotate than static grasps—but also reflects implicit assumptions about what robots need to learn. The emphasis on pick-and-place tasks in robotic manipulation has driven datasets toward stable grasping rather than dexterous manipulation. Dexonomy (Chen et al., 2025) attempts to bridge this gap by synthesizing 31 of the 33 GRASP types, demonstrating that systematic coverage is achievable through careful design. However, synthesis alone cannot capture the adaptive strategies humans employ when manipulating objects under uncertainty.

The taxonomy coverage problem extends beyond grasp types to manipulation primitives. While grasping receives extensive attention, equally important skills like controlled sliding, compliant interaction, and coordinated multi-finger motion remain understudied. These gaps in dataset coverage directly limit the capabilities of learned manipulation systems, creating a feedback loop where robots struggle with tasks that lack training data, reinforcing the focus on well-covered scenarios. Breaking this cycle requires datasets that systematically address the full spectrum of human manipulation capabilities, not just those easiest to capture or most immediately useful for current robotic systems.

## A.3 DATASET DETAILS

### A.3.1 LIST OF OBJECTS

| ID | Type | Size (cm) | Weight (g) | Placement | Schematic diagram |
|---|---|---|---|---|---|
| cube1 | Large cube | $8 \times 8 \times 8$ | 125 | – | |
| cube2 | Small cube | $5 \times 5 \times 5$ | 37 | – | |
| cuboid1 | Flat cuboid | $15 \times 15 \times 2$ | 128 | Large face down; side face down | |
| cuboid2 | Long cuboid | $15 \times 5 \times 2.5$ | 55 | Long edge along X-axis | |
| cuboid3 | Small cuboid | $5 \times 5 \times 2.5$ | 22 | – | |
| cylinder1 | Thick cylinder | $D7 \times 15$ | 140 | – | |
| cylinder2 | Disk cylinder | $D10 \times 2$ | 50 | – | |
| cylinder3 | Long cylinder | $D3 \times 10$ | 25 | – | |
| cylinder4 | Thin cylinder | $D1 \times 15$ | 8 | – | |
| cylinder5 | Mini cylinder | $D2 \times 2.7$ | 6 | – | |
| cylinder6 | Short thick cylinder | $D8 \times 10$ | 123 | – | |
| sphere1 | Large sphere | $D4$ | 13 | – | |
| sphere2 | Medium sphere | $D3$ | 8 | – | |
| sphere3 | Small sphere | $D2$ | 4 | – | |
| cube2H | Small cube (heavy) | $5 \times 5 \times 5$ | 123 | – | |
| cuboid2H | Long cuboid (heavy) | $15 \times 5 \times 2.5$ | 203 | Long edge along X-axis | |
| cuboid3H | Small cuboid (heavy) | $5 \times 5 \times 2.5$ | 82 | – | |
| cylinder1H | Thick cylinder (heavy) | $D7 \times 15$ | 554 | – | |
| cylinder4H | Thin cylinder (heavy) | $D1 \times 15$ | 14 | – | |
| cylinder5H | Mini cylinder (heavy) | $D2 \times 2.7$ | 10 | – | |
| cylinder6H | Short thick cylinder (heavy) | $D8 \times 10$ | 472 | – | |
| sphere1H | Large sphere (heavy) | $D4$ | 35 | – | |
| mayonnaisebottle | Mayonnaise bottle | – | 224 | Base down | |
| banana | Banana | – | 75 | Long edge along X-axis | |
| bowl | Bowl | – | 42 | Base down | |
| largeclamp | Clamp | – | 59 | Base down | |
| pitcherbase | Pitcher | – | 482 | Base down | |
| powerdrill | Power drill | – | 228 | Handle along Z-axis | |
| scissor | Scissors | – | 32 | Base down | |

Table 3: List of objects with dimensions, weights, placement setup, and schematic diagrams.

### A.3.2 CREATING MANO MODEL

**Estimate MANO shape** We derive per-subject MANO shape parameters (*betas*, 10D) using the HaMeR hand mesh regressor (Pavlakos et al., 2024). For each participant, we detect right hand, crop the corresponding RGB patches, and run HaMeR to obtain MANO parameter estimates (betas,

global orientation, and hand pose). We then aggregate the betas across valid crops by robust averaging (mean with outlier rejection) to produce a stable right shape vector per subject. During mocap fitting, these betas are kept fixed while we optimize only the global wrist transform and per-frame MANO pose to align markers. The resulting subject-specific shapes are reused across sessions and in simulation for retargeting.

**MANO fitting** *MANO* is a parametric hand model that maps a 10 dimensional shape vector $\beta$ and pose parameters $\theta$, together with a global wrist transform, to a differentiable hand mesh and joint locations via linear blend skinning. We parameterize each subject's hand with MANO. From a short calibration sequence we estimate a subject specific shape vector $\hat{\beta} \in \mathbb{R}^{10}$ and fix the marker to model correspondence. For every frame $t$ in a trial, given the measured 3D marker positions $\{x_{i,t}\}_{i=1}^N$, we solve for the global wrist transform $T_t \in SE(3)$ and the MANO pose parameters $\theta_t$ by minimizing the marker to model discrepancy:

$$\min_{T_t,\, \theta_t} \; \sum_{i=1}^{N} \left\| x_{i,t} - p_i\Big(T_t,\, \hat{\beta},\, \theta_t\Big) \right\|_2^2 \;+\; \lambda_{\text{pose}}\, \psi(\theta_t), \tag{3}$$

where $p_i(\cdot)$ returns the 3D location of the $i$th virtual marker on the MANO surface induced by $(\hat{\beta}, \theta_t)$ and transformed by $T_t$, and $\psi$ encodes soft joint limit priors. The optimization yields a time series of wrist rotation and translation in the world frame and local joint rotations for all MANO joints, together with the fixed subject shape $\hat{\beta}$. Given $(\hat{\beta}, \theta_t, T_t)$ we run the MANO forward model to obtain per frame joint locations and the hand mesh vertices, aligned to the mocap world frame and time synchronized with the object pose and RGB-D streams.

## A.4 DATA PROCESSING

**Pre-processing:** We first remove unusable segments (e.g., long occlusions or broken marker constellations) and flag anomalous trials with quality tags. For each trial, the object 6-DoF pose in the mocap world frame is recovered from its four reflective markers using rigid-body registration, yielding a sequence of rotations and translations. For the hand, we perform a subject-specific calibration to estimate the MANO shape vector and fix marker-to-bone offsets. The right-hand marker constellation is then tracked through time to obtain the wrist pose and absolute joint rotations in the mocap world frame.

To align the mocap output with the MANO coordinate system, we apply a fixed transformation $T_{MW}$ from the mocap world (W) to the MANO world (M). The resulting global and joint-level mappings are given by:

$$T_H^M(t) = T_{MW}\, T_H^W(t), \qquad T_O^M(t) = T_{MW}\, T_O^W(t), \qquad R_j^M(t) = R_{MW}\, R_j^W(t).$$

Here $T_H^W(t)$ and $T_O^W(t)$ denote the hand and object poses estimated from mocap, and $R_j^W(t)$ is the absolute rotation of joint $j$. These are transformed into MANO's world to produce consistent per-frame hand and object states. All streams are time-aligned with the front and side RGB-D cameras using recorded timestamps, and synchronization is verified with a short calibration gesture so that each RGB-D frame corresponds to a well-defined hand–object configuration.

## A.5 PHYSICS-BASED FORCE RECONSTRUCTION VIA RL

### A.5.1 ACTIONS

Table 4: Action space and DoF allocation. Actions are continuous and bounded.

| Group | Symbol | Dim | Meaning | Typical bounds |
|---|---|---|---|---|
| Wrist translation | $a_t^{\text{lin}}$ | 3 | Cartesian increments $(x, y, z)$ | $[-a_{\max},\, a_{\max}]$ m/step |
| Wrist rotation | $a_t^{\text{rot}}$ | 3 | Roll–pitch–yaw increments | $[-r_{\max},\, r_{\max}]$ rad/step |
| Finger joints | $a_t^{\text{fin}}$ | $n_f$ | Per-joint increments (thumb+4 fingers) | Joint-group specific caps |
| Total | | $6+n_f$ | | |

Our control scheme uses exponentially-weighted cumulative actions to ensure smooth manipulation trajectories. Rather than applying raw policy outputs directly, we maintain a smoothed action signal that prevents jittery movements while preserving responsiveness to policy decisions.

For each action component (wrist translation, rotation, finger joints), we compute the executed control signal $u_t$ as:

$$u_t = \tau \cdot u_{t-1} + a_t \tag{4}$$

where $a_t$ is the raw policy action and $\tau \in [0.8, 0.95]$ is the decay factor. This creates an exponential moving average where recent actions have higher weight: $u_t = \sum_{i=1}^{t} \tau^{t-i} a_i$.

**Example:** For wrist translation, if the policy outputs small incremental movements $a_t = [0.01, 0, 0]$ (1cm in x-direction), the executed control maintains momentum from previous steps while smoothly incorporating the new command. With $\tau = 0.9$, after 5 identical actions, the cumulative effect $u_5 \approx 0.041$ meters provides smooth acceleration rather than discrete jumps.

This smoothing is crucial for stable grasping: abrupt changes in wrist pose or finger positions can break contact or cause objects to slip, while our scheme maintains contact stability throughout complex manipulation sequences.

### A.5.2 OBSERVATIONS

Table 5: Observation (training-time privileged state). Replace $n_q, n_f$ with your exact values.

| Block | Symbol | Dim | Definition / Contents | Notes |
|---|---|---|---|---|
| Joint positions | $q_t$ | $n_q$ | Hand joint angles (local) | Normalized by joint limits |
| Joint velocities | $\dot{q}_t$ | $n_q$ | Finite-difference of $q_t$ | Clipped to safe range |
| Object pose | $T_{o,t}$ | 7 | $(p_{o,t} \in \mathbb{R}^3,\ R_{o,t} \in \mathbb{H}^1)$ | Position + quaternion |
| Hand–object relative | $\Delta_{ho,t}$ | 6 | $(p_{o,t} - p_{h,t},\ \text{relative ori})$ | Minimal relative pose |
| Target object pose | $\hat{T}_{o,t}$ | 7 | Mocap-derived target $(\hat{p}_{o,t}, \hat{R}_{o,t})$ | Reference trajectory |
| Short-horizon target | $\hat{p}_{o,t+\tau}$ | 3 | Future object position (e.g., $\tau{=}10$ steps) | Trajectory preview |
| Cumulative offsets | $\delta p_t$ | 3 | Position residual accumulator | For residual control |

### A.5.3 REWARDS

Table 6: Reward and penalty components used during physics-validated replay.

| Component | Symbol | Definition |
|---|---|---|
| Distance reward | $r_{\text{dist}}$ | Distance between sim and mocap object: $\exp(-60 \cdot \text{goal\_dist})$ |
| Rotation reward | $r_{\text{rot}}$ | Angular difference: $\exp(-10 \cdot \text{rot\_error})$ |
| Action penalty | $c_{\text{act}}$ | Cumulative action offset: $\|a_t\|_2^2$ |
| Total reward | - | $r_t = r_{\text{dist}} + r_{\text{rot}} - c_{\text{act}}$ |

