# OpenReview forum: "DexCanvas: Bridging Human Demonstrations and Robot Learning for Dexterous Manipulation"
_ICLR.cc/2026/Conference — ICLR 2026 Conference Desk Rejected Submission_

### Official Review · Reviewer_Mv9s · 2025-11-01

**Soundness:** 3
**Presentation:** 2
**Contribution:** 2
**Rating:** 4
**Confidence:** 4

**Summary:**

DexCanvas presents a 7,000-hour large-scale hybrid real-synthetic dataset of human dexterous manipulation, seeded from 70 hours of motion-captured demonstrations. The dataset is structured across 21 manipulation types (per the Cutkosky taxonomy) and augments each motion with physics-validated contact forces via a “real-to-sim RL pipeline” that trains actuated MANO hand policies to reproduce demonstrations in simulation, thereby recovering per-frame contact forces. This bridges gaps between pure mocap and synthetic data, providing the first large dataset combining multi-view RGB-D, precise kinematics, and physically consistent forces.

**Strengths:**

- The authors provided detailed instructions on data processing and synthesis. This can help future studies to reproduce and expand the dataset.
- The authors have explored training an RL policy that controls the MANO hand using the dataset.

**Weaknesses:**

1.  The “contact force extraction” pipeline relies entirely on simulated replay rather than physical measurement. Specifically, the authors train an RL policy to match the object trajectory observed in mocap, and then read the simulator’s internal contact forces as ground truth. This procedure amounts to a sim-to-real matching step, not genuine force recovery, since the resulting forces are determined by the simulator’s contact parameters rather than the real dynamics. The mapping from motion to force is underdetermined, and the paper does not provide validation showing that these forces correspond to reality (e.g., via tactile or instrumented measurements). Therefore, the claim of “physically validated force profiles” is overstated.

2. The trajectory-tracking policy employed in this work follows a standard imitation or RL-with-demonstrations setup. Similar approaches have been extensively explored, for example, in Rajeswaran et al. (2018, “Learning Complex Dexterous Manipulation with Deep Reinforcement Learning and Demonstrations”). It would strengthen the paper if the authors could demonstrate how established algorithms such as DAPG or related imitation-based RL methods could directly leverage the proposed dataset, rather than re-implementing a similar framework.

**Questions:**

Please see the weaknesses section. Thanks!

---

> ### Author Response · Authors · 2025-11-24
>
> We thank the reviewer for the thoughtful feedback and for pointing out important issues in how we present our contribution.
>
> We agree that the contact forces we obtain are strongly influenced by the simulator’s contact model and parameters, and that this dependence is not discussed in enough depth in the paper. Our forces are constrained to reproduce the observed object trajectories under the chosen simulator, but they are not validated against real-world force measurements. Thus, they should be understood as physically consistent within a given simulation model, not as true ground-truth forces.
>
> We also agree that the phrase “physically validated force profiles” is too strong given the current level of validation. Our intention was to emphasize consistency with simulated physics and real trajectories, not to claim experimental validation in the real world. Furthermore, we recognize that our RL formulation is closely related to existing demonstration-guided RL methods such as DAPG, and that the novelty of this work does not lie in the RL algorithm itself, but in using such a pipeline to build a large-scale contact-rich dataset.
>
> Taken together, these comments highlight that the current submission under-explains in some aspects. We accept that the paper in its present form is not yet fully convincing and will need substantial reworking to clarify its claims and limitations.

---

### Official Review · Reviewer_5HJA · 2025-11-01

**Soundness:** 2
**Presentation:** 3
**Contribution:** 3
**Rating:** 2
**Confidence:** 4

**Summary:**

This paper introduces DexCanvas, a large-scale hybrid real-synthetic dataset for dexterous manipulation, constructed from 70 hours of real human demonstrations and expanded to 7,000 hours via physics simulation. The key innovation lies in a real-to-sim pipeline that uses reinforcement learning to reproduce human motions in simulation, thereby extracting physically consistent contact forces and object dynamics that are missing in motion capture data. The dataset is systematically organized using a manipulation taxonomy and provides rich multimodal annotations, including MANO parameters, RGB-D, and force profiles, enabling research in contact-rich robotic control and cross-morphology skill transfer.

**Strengths:**

The paper presents a novel real-to-sim pipeline using reinforcement learning to generate scalable, high-quality contact force annotations from human demonstrations, offering a cost-effective alternative to instrumented hardware. The dataset is rich in multimodal sensory signals, systematically organized by manipulation taxonomy, and provides rare physical annotations that are valuable for dexterous manipulation research.

**Weaknesses:**

The DexCanvas dataset makes a valuable contribution by bridging human demonstrations with physics-based robotic learning, and the proposed RL-based real-to-sim pipeline for extracting contact forces is both innovative and promising for scalable data generation. However, I have some major concerns.

First, the work is very similar to ManipTrans[1], which retargets human trajectories and objects be manipulated scanned for both single and two hands to physical plausible demonstration data by RL and also generates large scale manipulation trajectories.

Second, although the authors conduct statistical analysis about the proposed dataset, its value for the target tasks are not demonstrated. Baseline and benchmark with the dataset compared with prior works without the dataset or using other datasets on several downstream tasks are required to consolidate the work.

[1] CVPR2025：MANIPTRANS: Efficient Dexterous Bimanual Manipulation Transfer via Residual Learning
]

In addition, I have a few moderate concerns.

First, the objects used are 3D-printed with embedded markers, which may differ from real-world objects in terms of material properties, surface textures, mass distribution, and shape variation—potentially affecting the realism and generalization of policies trained on this data.

Second, while the dataset is expanded to 7,000 hours via simulation, the real demonstrations are collected on only 30 objects, and although perturbations are applied, the diversity of object geometries and manipulation contexts could be limited.

Third, overclaim of the contribution of the dataset. In Table 1, the manipscope of the work is general. However, in fact, the current dataset focuses exclusively on single-handed manipulation, leaving bimanual coordination unexplored.

Overall, the dataset is well-constructed and offers potential for advancing dexterous manipulation research. However, given the limited contribution compared with prior work and the missing of the target tasks evaluation, I suggest borderline reject and the authors to further refine the work.

**Questions:**

see weakness

---

> ### Author Response · Authors · 2025-11-24
>
> We thank the reviewer for carefully reading the paper and for the candid comments on novelty, scope, and practical value.
>
> We acknowledge that there is clear conceptual overlap between our work and ManipTrans, since both use human hand data and simulation for dexterous manipulation. Our intention was not to claim exclusivity over this idea. DexCanvas is primarily focused on providing a large-scale, systematically organized single-hand manipulation dataset with multimodal annotations (including forces), whereas ManipTrans focuses on a specific retargeting method for bimanual demonstrations. However, we recognize that the current paper does not clearly articulate this distinction and that, as written, our positioning relative to ManipTrans and similar work is insufficient.
>
> We also agree that the practical value for downstream tasks is not convincingly demonstrated in this submission. We mainly validate the real-to-sim reconstruction and provide dataset statistics, but we do not yet present strong robot control or policy-learning results that build on DexCanvas.
>
> Overall, we accept that this version of the work is not yet mature: the connections to closely related work are not explained clearly enough, and the empirical validation is incomplete. We appreciate the reviewer’s comments and will take them into account when we substantially rework and extend this line of research.

---

### Official Review · Reviewer_D3V2 · 2025-11-01

**Soundness:** 2
**Presentation:** 2
**Contribution:** 3
**Rating:** 4
**Confidence:** 4

**Summary:**

This paper presents a large-scale dataset of 7,000 hours simulated  human dexterous manipulation derived from 70 hours of mocap-based demonstrations with 21 manipulation types. The real-to-sim pipeline trains actuated MANO-hand controllers in simulation to reproduce human object trajectories, offering synchronized RGB-D, MANO parameters, and contact annotations. This dataset might inspire research in contact-rich robot learning, control, and cross-morphology skill transfer

**Strengths:**

Integrates real human demonstrations with physics-based rollouts -- using physics tracking policy to clean data is an emerging field.

Uses a 22-camera mocap system synchronized with RGB-D sensors and precisely aligned 3D-printed objects for accurate geometry and motion data

**Weaknesses:**

Although the paper acknowledges several limitations, (1) despite its larger scale, it employs only a limited set of primitive objects; (2) its policy is based on MANO rather than a robot hand, requiring additional retargeting for robot learning that inevitably degrades data quality; and (3) it adopts a per-object policy that lacks scalability and generalization to unseen clean data. These issues have already been addressed by many RL-based studies, including but not limited to DexTrack [1], ManipTrans [2], and Dexplore [3]. Given these existing works, the paper fails to demonstrate advantages over such limitations, which constitutes a critical weakness.

[1] Liu, Xueyi, et al. "Dextrack: Towards generalizable neural tracking control for dexterous manipulation from human references." arXiv preprint arXiv:2502.09614 (2025).

[2] Li, Kailin, et al. "Maniptrans: Efficient dexterous bimanual manipulation transfer via residual learning." Proceedings of the Computer Vision and Pattern Recognition Conference. 2025.

[3] Xu, Sirui, et al. "Dexplore: Scalable Neural Control for Dexterous Manipulation from Reference Scoped Exploration." Conference on Robot Learning. PMLR, 2025.

Such limitations are also partially due to no downstream learning evaluation (e.g., trainingBC/diffusion/VLA models on DexCanvas vs. prior datasets) nor real‑robot transfer to establish practical value at deployment time; **As explicitly claimed in the supplementary, the authors defer these important experiments to rebuttal**, I am not sure if it is an encouraged behavior for ICLR


Success and force read‑outs can be sensitive to different physical parameters. The paper lacks ablation or sensitivity studies to demonstrate robustness of recovered forces across reasonable physics parameter ranges.

Although the dataset is large, much of its scale stems from repeatedly rolling out the same simulation policy with objects placed in slightly varied initial positions. As a result, the data diversity primarily reflects pose perturbations rather than behavioral or strategy diversity. It remains unclear how the augmented dataset captures genuinely distinct manipulation behaviors to benefit the robot learning side

**Questions:**

For the in‑hand category or tasks requiring precise manipulation, what would be the orientation tracking, contact persistence, or slip statistics in addition to positional success with fixed threshold

---

> ### Author Response · Authors · 2025-11-24
>
> We thank the reviewer for the constructive and balanced feedback.
>
> We agree that our current experiments do not sufficiently demonstrate advantages over existing datasets (such as GRAB or ARCTIC) for downstream learning. The evaluation in this paper is largely limited to showing that our real-to-sim pipeline can track demonstrations and to describing the dataset structure. This is clearly not enough to fully justify the impact of DexCanvas, and more thorough downstream comparisons and policy-learning results are needed beyond this submission.
>
> Regarding the use of MANO instead of a robot hand, our goal is to build a human-hand–centric, morphology-agnostic dataset that can in principle be retargeted to different robot hands. However, we acknowledge that we do not provide robot retargeting or real-robot experiments in this paper, and that this reduces the demonstrated practical value of the work. Similarly, the choice to train separate policies per object–manipulation pair is an engineering compromise rather than a principled solution, and we agree that it raises scalability concerns that we do not resolve here.
>
> We also agree that the paper lacks ablation and sensitivity analysis for physical parameters (mass, friction, stiffness), and that part of the diversity in the simulated data comes mainly from perturbations around demonstrations rather than fundamentally different strategies. Taken together, these points show that the current version of the work is still incomplete. We appreciate these comments and will use them to guide a more thorough reworking of the paper.

---

### Official Review · Reviewer_osbF · 2025-11-03

**Soundness:** 3
**Presentation:** 3
**Contribution:** 2
**Rating:** 2
**Confidence:** 4

**Summary:**

This work introduces DexCanvas, a large dataset for manipulation with dexterous hands, by bridging mo-cap human demonstrations with learning. It is prepared by using 70 hours of real human manipulation data using multi-modal capture (RGB-D and Vicon-style motion capture) spanning 21 types of manipulation. The mo-cap data has some systematic errors when replayed open-loop in sim. Hence, the work uses RL (PPO) to make these simulated manipulation demos physically consistent; the RL agent adds a residual action to the mocap data to track the original demos trajectories of hand and objects. Once they have such data, they record attributes like contact points and forces from the sim to produce a richer demonstration data than the original mo-cap data from the real world. This creates the original 70 hours of demo data with physics-aware contact information in sim. Then, they add perturbations to the environment to create a larger set of rollouts, which gives you 100x the training size -- going to 7000 hours of demo data. The authors demonstrate this pipeline that effectively reconstructs natural human motion, creates physics-aware demo data, and provides fine-grained force profiles for various grasping strategies. This is great work that is well motivated. But there are no tests to validate the hypothesis of the dataset.

An alternative story for the paper could be the _pipeline_ that allows for such physics-aware data collection. You start with some mocap data, and explain the pipeline (& RL) to move it to sim and record physically consistent motion, contact, and pose data -- which gives you a much richer dataset, for a much richer manipulator. The output of this paper could be that pipeline, and a subsequent paper could be the actual dataset, after it is appropriately validated.

**Strengths:**

* Well motivated. Good comparison with current dexterous hand data that's out there. We need more robotics data, especially for hands that are not parallel jaw grippers. Crucially, we need contact-informed data, not just kinematics data.

* Effort has gone into capturing different types of manipulations, getting MANO parameters like the shape params (beta), that account for hand pose beyond just joint angles in the fingers.

**Weaknesses:**

1. The whole paper is a dataset -- but there is no downstream use of the dataset to validate its effectiveness. The authors state that _"Additional experiments demonstrating cross-dataset applicability and downstream task evaluation will be provided during the rebuttal phase."_ They are already baking in work for the rebuttal before the reviewers ask for any clarifications or expts. I am not sure if that's what the rebuttal phase is for. Reviewers won't get enough time to understand the new material you already plan to add for the rebuttal (architectures, training pipelines, everything else that is done after the dataset is created).
1. The effectiveness of a trained policy is tested - but that is the policy trained to generate the dataset. There is no test to demonstrate the effectiveness of the dataset.
1. Single RL policy per object-manipulation pair: A single policy could benefit from cross-task transfer. This is already mentioned as a limitation in the paper.
1. Contact-rich forces are being recorded in sim (IsaacGym). But simulation is not the best for accurate contact-rich data. And aspects like object deformation is not captured. Papers like [1] show how we need a good contact model for things like USB-insertion, and truly show the limits of contact-rich simulation. Hence, this simulated data will like be a supplementary add-on; the main demos would be real human demos with true force information to serve as the ground truth.

[1] "Efficient Online Learning of Contact Force Models for Connector Insertion" https://arxiv.org/pdf/2312.09190

**Questions:**

See weaknesses section.

---

> ### Author Response · Authors · 2025-11-24
>
> We thank the reviewer for the detailed and thoughtful comments.
>
> We agree that the main weakness of this submission is the lack of downstream task evaluation that clearly demonstrates the practical value of DexCanvas. In the current paper, our experiments focus on validating the real-to-sim tracking pipeline and showing that we can reconstruct physically consistent trajectories and forces, but they do not provide convincing evidence for policy learning or other downstream tasks. We acknowledge that this makes the submission incomplete and that more extensive downstream experiments (e.g., policy learning on DexCanvas vs. existing datasets) are necessary when we further develop this line of work.
> We also agree that training one RL policy per object–manipulation pair is not scalable. This is an implementation choice made to ensure robust tracking, not a conceptual requirement of the approach. We will further develop this work to make sure it is more scalable.
>
> Regarding contact forces from simulation, we acknowledge that they are not real-world ground truth. Our intention is to provide forces that are physically consistent with the observed object motion under a given simulator, not to claim exact real-world forces. We also recognize that direct force measurements on real objects would be a stronger reference, even though they are difficult to obtain at our scale. Overall, we accept that the current submission does not fully address these concerns and that the work needs further refinement.

---

### Author Response · Authors · 2025-11-24
**Summary of our response**

We sincerely thank all reviewers for carefully reading our submission “DexCanvas: Bridging Human Demonstrations and Robot Learning for Dexterous Manipulation” and for the thoughtful and constructive comments.

Overall, we fully acknowledge that the current version of the paper is not yet as mature as it should be, especially in terms of:
the experimental section (limited baselines, ablations, and downstream tasks)

At the same time, we are grateful for the recognition of the core idea: using a hybrid real-to-sim pipeline to turn human demonstrations into a large-scale, force-aware dexterous manipulation dataset, organized by a manipulation taxonomy and targeted at learning robot control.

Given the points raised in the reviews, rather than making only minor revisions, we plan to substantially rework the paper—including stronger experiments, clearer analysis, and a more careful comparison with related work before submitting to a future venue.

---

### Note · Program_Chairs · 2026-01-17
**Submission Desk Rejected by Program Chairs**

The following references in this submission do not refer to real documents and/or have major errors in bibliographic information:

 Yifei Zheng, Yan Wang, Andrea Wetzler, and Pascal Fua. Hi4d: 4d instance segmentation of close human interaction. In IEEE/CVF Conference on Computer Vision and Pattern Recognition (CVPR), pp. 17011-17021, 2023.